# GUARANTEEING CONSERVATION OF INTEGRALS WITH PROJECTION IN PHYSICS-INFORMED NEURAL NETWORKS

## ABSTRACT

We propose a novel projection method that guarantees the conservation of integral quantities in Physics-Informed Neural Networks (PINNs) Raissi et al. (2019). While the soft constraint PINNs use to enforce the structure of partial differential equations (PDEs) enables necessary flexibility during training, it also permits the discovered solution to violate physical laws. To address this, we introduce a projection method that guarantees the conservation of the linear and quadratic integrals, both separately and jointly. We derived the projection formulae by solving constrained non-linear optimization problems and found that our PINN modified with the projection, which we call PINN-Proj, reduced the error in the conservation of these quantities by three to four orders of magnitude compared to the soft constraint and marginally reduced the error in the PDE solution. We also found evidence that the projection improved convergence through improving the conditioning of the loss landscape. Our method holds promise as a general framework to guarantee the conservation of any integral quantity in a PINN if a projection is found and is tractable in neural network training.

## 1 INTRODUCTION

Recently, neural network-based approaches have emerged as powerful methods to solve and model PDEs. Among these are Physics-Informed Neural Networks (PINNs) Raissi et al. (2019), which can parameterize the solution of a partial differential equation (PDE) with little to no training data and without requiring a mesh. The structure of the PDE and the initial and boundary conditions of the solution are enforced by the loss function. This creates a soft constraint on the physics equation, where the PINN is penalized for predictions that deviate from it. The soft constraint allows flexibility during training so that the PINN learns a solution that fits both the training data and the governing PDE.

This soft constraint, however, permits the solution learned by the PINN to violate conservation laws, which are fundamental principles of physics. If PINNs learned solutions that exactly obeyed conservation laws, they would be more physically accurate and more trustworthy when applied to model real-world physical systems. We cannot exactly enforce conservation laws using another soft constraint, as it could still be violated in a valid solution. Instead, we must impose a hard constraint that ensures that the PINN always obeys the laws of conservation regardless of the nature of the data. The conservation laws that we focus on are in the form of integral quantities that must remain constant or evolve according to changes in the solution domain.

In this work, we propose a novel framework that can guarantee the conservation of some integral quantity by projecting the solution prediction of the PINN into a functional space where the conservation is not violated. This can be done by framing the projection as a constrained non-linear optimization problem where we are minimizing the distance between the solution and the hyperplane where the integral is conserved. We then use Lagrange multipliers to find the unique solution to this problem. Given that a projection that is differentiable and tractable in training is found, it will ensure conservation exactly. We demonstrate that this projection method guarantees global conservation of the linear and quadratic integral, both separately and at the same time and on conserved

and non-conserved systems. We define *guarantee* as ensuring that the only sources of error are from the data and machine precision and not from the method.

We refer to our PINN model modified with the projection method as PINN-Proj. We compare its performance to an unmodified PINN and a PINN model modified with a soft constraint on the conservation of the same quantity or quantities, referred to as PINN-SC. The models were trained on small samples of PDE solutions and evaluated on the error of their PDE solution predictions $u$ and the conservation of the chosen integral quantity. We refer to these integral quantities as *conserved quantities c*. Our main contributions in this paper are as follows:

- We propose a novel projection method and show that it guarantees the conservation of the linear integral, quadratic integral, and both simultaneously

- We derive the projections for our conserved quantities by solving non-linear constrained optimization problems

- We show that PINN-Proj successfully guarantees the conservation of the integral quantities while marginally improving the accuracy of the PDE solution

- We analyze the convergence and loss landscape of PINN and PINN-Proj using spectral analysis of the Hessian of the loss

## 2 RELATED WORK

**Soft Constraints on PINNs**  A soft constraint can be used to enforce the conservation law in a PINN by adding a loss term that penalizes the difference between the conserved quantity in the current prediction and the ground truth. Training can either be done entirely with this modified loss function Fang et al. (2022), or in two stages where the modified loss function is used in a second stage of training Lin & Chen (2022). Creating soft constraints in this way can also be used to enforce conservation of flux between two neighboring discrete subdomains Jagtap et al. (2020).

**Hard Constraints on PINNs**  Hard constraints have been used in PINNs to strictly enforce initial and boundary conditions Xiao et al. (2024). This was done by modifying the PINN solution with a distance function that vanishes at the boundaries and a specific solution that automatically satisfies the boundary and initial conditions. KKT-hPINN uses KKT conditions, which describe optimality conditions in constrained optimization, to create an untrainable projection on the output of a PINN that conserves a desired quantity, but is limited to linear constraints due to relying on the closed-form analytical solution of a linear system to create the projection Chen et al. (2024). Projection on the output of a PINN has also been used to guarantee conservation by applying multiple Newton iterations after each training step to move the solution back into the conservation manifold Cardoso-Bihlo & Bihlo (2025). This projection method, however, relies on making iterative soft updates on the PINN output and does not incorporate the conservation constraint directly as a differentiable layer into the PINN. Numerical error values are also not reported, so it is difficult to quantify the performance of this hard constraint.

**Exactly Learning Conservation Laws**  Conservation laws can be directly incorporated into the structure of neural networks to create hard constraints as well. The Hamiltonian Greydanus et al. (2019) or Lagrangian Cranmer et al. (2019), which describe how total energy is conserved in a system, can be parametrized by a neural network so that the quantities are always conserved in any solution it learns. Similarly, hard constraints can also be created by construction in a neural network on conservation laws by parameterizing a divergence-free vector field that corresponds to the continuity equation Richter-Powell et al. (2022). Due to the divergence-free assumption, however, these neural conservation laws are limited to forms of conserved systems, whereas PINNs can solve and model non-conservative systems that have sources, sinks, or boundary flux. PINNs are also not limited to PDEs that are forms of the continuity equation. Enforcing conservation using the continuity equation modeled as a divergence-free field has also been applied to neural operators, which instead learn the governing PDE directly from solution data Liu et al. (2023). A differential projection layer can also be incorporated after a neural network to find the optimal linear combinations of basis functions to learn the governing PDE and enforce it as a hard constraint Négiar et al. (2022).

## 3 METHOD

### 3.1 PHYSICS-INFORMED NEURAL NETWORK

We define the general form of a partial differential equation (PDE) as

$$0 = u_t + \mathcal{N}[u], \tag{1}$$

where $u(x, t)$ is the solution and a function of spatial coordinate $x$ and time $t$. $\mathcal{N}[\cdot]$ is a differentiable non-linear operator that acts on $u(x, t)$. The physics-informed neural network (PINN) is defined as

$$f := u_t + \mathcal{N}[u], \tag{2}$$

where $f(x, t)$ is a function and the residual between $u_t$ and $\mathcal{N}[u]$. The PINN learns a parameterized solution $u_\theta(x, t)$ when $\mathcal{N}[\cdot]$ is known by using the loss function, $\mathcal{L}$, defined as

$$\mathcal{L} = \frac{1}{N_u} \sum_{i=1}^{N_u} |u_\theta(x_u^i, t_u^i) - u^i|^2 + \frac{1}{N_f} \sum_{i=1}^{N_f} |f(x_f^i, t_f^i)|^2. \tag{3}$$

Here, $\{x_u^i, t_u^i, u^i\}$ are the initial and boundary data on $u(x, t)$, while $\{x_f^i, t_f^i\}$ are collocation points randomly sampled from the domain independently. The first term of $\mathcal{L}$ is the mean squared error between the PINN prediction $u_\theta(x, t)$ and ground truth $u^i$. The second term of $\mathcal{L}$ is the mean squared error of $f(x, t)$, as the residual of the PDE is dependent on the learned $u_\theta(x, t)$ and is penalized to enforce the PDE's structure on the parameterized solution.

### 3.2 PHYSICS-INFORMED NEURAL NETWORK WITH SOFT CONSTRAINT

We include PINN-SC to act as a benchmark for a soft constraint used to enforce conservation. This constraint was incorporated by adding a term to the loss function that penalized the difference between the predicted conserved quantity $\hat{c}$ and ground truth $c$

$$\mathcal{L}_{SC} = \mathcal{L} + \lambda \frac{1}{T} \sum_{t=1}^{T} |c(t) - \hat{c}(t)|^2. \tag{4}$$

We found that adding a weight term $\lambda$ to the conservation loss improved the performance of PINN-SC, establishing it as a more rigorous benchmark.

### 3.3 CONSERVATION LAWS

Many PDEs' governing physical systems are naturally expressed as conservation laws. A canonical form of such equation is

$$\frac{\partial u}{\partial t} + \nabla \cdot \mathbf{F}(u) = 0, \tag{5}$$

where the flux $\mathbf{F}(u)$ denotes the flow of a quantity $u$ across a given spatial domain. This equation illustrates that the change in the conserved quantity over time is related to the divergence of its flux. To analyze the global behavior, we integrate the conservation law over a spatial domain $\mathcal{X}$,

$$\frac{d}{dt} \int_{\mathcal{X}} u \, dx = -\int_{\mathcal{X}} \nabla \cdot \mathbf{F}(u) \, dx = -\oint_{\partial \mathcal{X}} \mathbf{F}(u) \cdot d\mathbf{S}. \tag{6}$$

The second equality is from applying the divergence theorem, which relates the volume integral of the divergence of a vector field to a surface integral over the boundary of the domain. This formulation identifies how the net change of the conserved quantity only depends on the flux across the boundary. If the flux through the boundary is zero, then the conserved quantity remains constant within the domain $\mathcal{X}$. If there exists flux across the boundary, the changes in the conserved quantity can be explicitly calculated,

$$c(t) = \int_{\mathcal{X}} u(x, t) \, dx, \tag{7}$$

where $\mathcal{X}$ represents the computational domain and $c(t)$ denotes the known total conserved quantity. The changes in the conserved quantity can also be calculated if there is a source or sink term in the domain. Under periodic, reflective, or homogeneous Neumann boundary conditions, $c(t)$ remains constant. Under other prescribed boundary conditions, $c(t)$ can be determined through temporal integration.

### 3.4 CONSERVATION OF THE LINEAR INTEGRAL

We define the linear integral of the parameterized solution of the PDE, $u_\theta(x, t)$ over the spatial domain $\mathcal{X}$ as

$$c(t) = \int_{\mathcal{X}} u_\theta(x, t) \, dx. \tag{8}$$

If we select a PDE where $u$ describes the velocity of a fluid, then this conserved quantity can be physically interpreted as the total momentum of the system.

### 3.5 CONSERVATION OF THE QUADRATIC INTEGRAL

We define the quadratic integral of the parameterized solution of the PDE, $u_\theta(x, t)$ over the spatial domain $\mathcal{X}$ as

$$c(t) = \int_{\mathcal{X}} u_\theta(x, t)^2 \, dx. \tag{9}$$

If we select a PDE where $u$ describes the velocity of a fluid, then this conserved quantity can be physically interpreted as total energy of the system.

### 3.6 PROJECTION METHOD FOR CONSERVATION OF THE LINEAR INTEGRAL

Because these integral constraints are intractable under a discretized PDE solution and neural network parameterized output $u_\theta(x, t)$, we instead approximate the integral with discretization and numerical integration over the neural network outputs,

$$\hat{c}(t) = \int_{\mathcal{X}} u_\theta(x, t) \, dx \approx \sum_{i=1}^{n} u_\theta(x_i, t)\Delta x = \mathbf{1}_n^\top u_\theta(x, t)\Delta x, \tag{10}$$

where $\mathbf{1}_n$ is the all-ones vector, $\Delta x$ represents the uniform discretization size, and $n$ is the total number of discretization points. To force the neural network output to respect the linear integral constraint, we consider the following optimization problem

$$\min_y ||\bar{u}_\theta(x, t) - y||^2 \ \text{ s.t. } \Delta x \mathbf{1}_n^\top y = c(t), \tag{11}$$

where $\bar{u}_\theta(x, t) = [u_\theta(x_1, t), u_\theta(x_2, t), ...u_\theta(x_n, t)]^\top$ is a vector that represents the discretized ground truth values of $u_\theta(x, t)$ arranged as a column vector, and $y$ is our prediction from our projection method. We solve this constrained optimization problem by Lagrange multiplier. This problem has unique global minimum because, geometrically, we are finding the point on a hyperplane that has the smallest distance from a given point. We find the solution to be

$$\tilde{u}_\theta(x, t) = u_\theta(x, t) + \frac{c(t)}{n\Delta x} - \sum_{i=1}^{n} \frac{\bar{u}_\theta(x_i, t)}{n} \tag{12}$$

where $u_\theta(x, t)$ is the unprojected output of the neural network and $\tilde{u}_\theta(x, t)$ is the projected output of the neural network.

### 3.7 PROJECTION METHOD FOR CONSERVATION OF THE QUADRATIC INTEGRAL

We can now use the same method to find a projection for a physical system that is governed by quadratic summation constraints. The conservation principle can be expressed as

$$\hat{c}(t) = \int_{\mathcal{X}} u_\theta(x, t)^2 \, dx \approx \sum_{i=1}^{n} u_\theta(x_i, t)^2 \Delta x = y^\top y \Delta x. \tag{13}$$

We then consider the following optimization problem,

$$\min_y ||\bar{u}_\theta(x, t) - y||^2, \ \text{ s.t. } \Delta x y^\top y = c(t), \tag{14}$$

and find the solution to be

$$\tilde{u}_\theta(x, t) = u_\theta(x, t) \sqrt{\frac{c(t)}{\sqrt{\sum_{i=1}^{n} \bar{u}_\theta(x_i, t)^2}}}. \tag{15}$$

### 3.8 COMBINED PROJECTION METHOD FOR LINEAR AND QUADRATIC INTEGRAL

For a physical system that is governed by both linear and quadratic summation constraints, we can also find a conservation principle that obeys both constraints. We apply both previously incorporated constraints in the same manner as before,

$$\min_y ||\bar{u}_\theta(x,t) - y||^2 \text{ s.t. } \Delta x y^\top y = c_1(t), \ \Delta x \mathbf{1}_n^\top y = c_2(t) \tag{16}$$

and find the solution to be

$$\tilde{u}_\theta(x,t) = (u_\theta(x_i,t) - \sum_{i=1}^n \frac{u_\theta(x_i,t)}{n})\sqrt{\frac{c_1(t)/\Delta x - c_2^2(t)/(n\Delta x^2)}{\sum_{i=1}^n (u_\theta(x_i,t) - \sum_{i=1}^n \frac{u_\theta(x_i,t)}{n})^2}} + \frac{c_2(t)}{n\Delta x} \tag{17}$$

The complete derivations of all projections can be found in the Appendix.

### 3.9 PROJECTION METHOD FOR NON-CONSERVED SYSTEMS

Our projection method allows $c(t)$ to either be constant, as in a conserved system, or changing, as in a non-conserved system where there is boundary flux or the addition or removal of mass. In the latter case, the projection method requires an additional step to fully characterize $c(t)$. $c(t)$ is calculated from a discretized PDE solution $u(x,t)$, so $c(t)$ is only defined at the $t$ values in this solution. However, since the collocation points used to calculate $f(x,t)$ are generated using Latin Hypercube Sampling over the computational domain, some of these $t$ values will not have defined $c(t)$ values. This does not affect a conserved system, as $c(t)$ is assumed to be the same for all times. However, for a non-conserved system, we estimate these missing $c(t)$ values by linear interpolation using the second-order accurate gradient from the closest defined $c(t)$ value.

### 3.10 PROJECTION METHOD FOR 2D SYSTEMS

We can also extend the projection method to two dimensional PDEs by integrating over both spatial dimensions in the calculation of $c(t)$. For spatial dimensions $x$ and $y$, the linear integral is

$$\hat{c}(t) = \int_{\mathcal{Y}} \int_{\mathcal{X}} u_\theta(x,y,t) \, dx \, dy \approx \sum_{j=1}^n \sum_{i=1}^n u_\theta(x_i,y_j,t)\Delta x\Delta y, \tag{18}$$

and we use a similar formula for other conserved quantities. We can extend the projection method into an arbitrary number of dimensions by integrating over all spatial dimensions of the solution to find the conserved quantity.

### 3.11 EXPERIMENTAL SETUP

We trained three different models for evaluation: PINN, PINN-SC, and PINN-Proj. The training setup for all models is the same as the original PINN method Raissi et al. (2019) unless otherwise mentioned. We used four PDE datasets generated from the Reaction-Diffusion Equation, Advection Equation, Wave Equation, and Korteweg-De Vries Equation. These were selected to include PDEs of different orders and applications and because they all conserve both the linear and quadratic integral (with the exception of Reaction-Diffusion). All PDEs were one-dimensional unless otherwise noted.

The computational domain for all PDEs was discretized into 256 spatial points and 100 temporal points. All solutions were generated using various FDM solvers and had homogeneous Neumann boundary conditions. The Advection Equation, Wave Equation, and Korteweg-De Vries Equation solutions were conserved systems with no net change in mass. The Reaction-Diffusion Equation included a reaction term that added mass to the system over time, so the modified projection method for non-conserved systems was used in this case.

The training set contained 100 points for the PDEs and were randomly sampled from each PDE dataset, and contained 10,000 collocation points that were generated using Latin Hypercube Sampling. Each test set was composed of all spatial and temporal points in the domain to assess the PINN's ability to model the full PDE solution. Training was done until the convergence criterion of the L-BFGS optimizer was reached. More details can be found in the Appendix.

### 3.12 EVALUATION

The models were evaluated on the error in predicting the PDE solution, called Error $u$, and on the error in the conservation of the conserved quantity, called Error $c$, in the solution. Error $c_L$ refers to the error of conservation of the linear integral and Error $c_Q$ refers to the error of conservation of the quadratic integral. Relative $\mathcal{L}_2$ error was used for Error $u$, and absolute $\mathcal{L}_1$ error was used for Error $c$. We also report the duration and number of epochs of training. We averaged all values over 10 trials.

## 4 RESULTS

| PDE | PINN | | |
|---|---|---|---|
| | Error $u$ | Error $c_L$ | Error $c_Q$ |
| Adv. Eq. | 1.58E-02 | 1.31E-01 | 1.70E-01 |
| Wave Eq. | 3.27E-02 | 2.10E+00 | 3.56E+00 |
| KdV Eq. | 6.22E-02 | 9.40E-01 | 1.37E+00 |
| React-Diff Eq. | 1.95E-03 | 1.34E-01 | 2.29E-01 |

Table 1: Average error for solution prediction (Error $u$) and conservation (Error $c_L$ and Error $c_Q$) for PINN

| PDE | PINN-SC | | | | | | | | |
|---|---|---|---|---|---|---|---|---|---|
| | Linear | | Quadratic | | Both | | | | |
| | Error $u$ | Error $c_L$ | Error $u$ | Error $c_Q$ | Error $u$ | Error $c_L$ | Error $c_Q$ | | |
| Adv. Eq. | 1.05E-02 | 3.01E-03 | 6.95E-03 | 1.64E-03 | **7.53E-03** | 3.46E-03 | 2.74E-03 | | |
| Wave Eq. | 1.97E-02 | 3.99E-01 | 4.29E-02 | 6.88E-01 | 3.55E-02 | 3.59E-01 | 7.20E-01 | | |
| KdV Eq. | **5.74E-02** | 5.70E-01 | **5.59E-02** | 6.25E-01 | **5.60E-02** | 3.63E-01 | 5.58E-01 | | |
| React-Diff Eq. | 2.09E-03 | 5.03E-02 | 2.31E-03 | 3.69E-02 | 2.09E-03 | 4.75E-02 | 4.70E-02 | | |

Table 2: Average error for solution prediction (Error $u$) and conservation (Error $c_L$ and Error $c_Q$) for PINN-SC

| PDE | PINN-Proj | | | | | | | | |
|---|---|---|---|---|---|---|---|---|---|
| | Linear | | Quadratic | | Both | | | | |
| | Error $u$ | Error $c_L$ | Error $u$ | Error $c_Q$ | Error $u$ | Error $c_L$ | Error $c_Q$ | | |
| Adv. Eq. | **8.26E-03** | **7.01E-06** | **4.88E-03** | **3.73E-06** | 9.96E-03 | **2.43E-06** | **1.30E-06** | | |
| Wave Eq. | **1.95E-02** | **1.06E-05** | 3.26E-02 | **1.02E-05** | 2.68E-02 | **1.84E-05** | **2.53E-05** | | |
| KdV Eq. | 5.93E-02 | **2.10E-05** | 5.74E-02 | **2.48E-05** | 5.96E-02 | **1.62E-04** | **2.47E-04** | | |
| React-Diff Eq. | **1.21E-03** | **1.10E-05** | 1.88E-03 | **1.75E-05** | 2.06E-03 | **3.17E-05** | **3.65E-05** | | |

Table 3: Average error for solution prediction (Error $u$) and conservation (Error $c_L$ and Error $c_Q$) for PINN-Proj

Table 1 shows the error on solution prediction and conservation for PINN, Table 2 shows the error on solution prediction and conservation for PINN-SC, and Table 3 shows the error on solution prediction and conservation for PINN-Proj. Values in Table 2 and Table 3 are bolded if they were the lowest error for that configuration. The relative performance of the models was very similar across the three settings. On solution prediction, PINN-Proj achieved the lowest Error $u$ on 11 of the 15 trials, and PINN-SC achieved the lowest Error $u$ on four of the 15 trials. Of these four trials, three were from the Korteweg-De Vries Equation and one was from the Advection Equation. It is notable, however, that the margin between Error $u$ values for all models was always less than one order of magnitude. Regarding conservation, PINN-Proj achieved the lowest Error $c_L$ and Error $c_Q$ on all

| PDE | Metric | PINN | PINN-SC | PINN-Proj |
|---|---|---|---|---|
| Advection Eq. | Duration (s) | 1.75 | 17.37 | 13.79 |
| | Epochs | 391 | 597 | 331 |
| Wave Eq. | Duration (s) | 2.27 | 20.47 | 20.32 |
| | Epochs | 339 | 633 | 435 |
| Korteweg-De Vries Eq. | Duration (s) | 36.20 | 57.96 | 82.88 |
| | Epochs | 4065 | 1760 | 1754 |
| Reaction-Diffusion Eq. | Duration (s) | 9.33 | 66.17 | 68.69 |
| | Epochs | 1710 | 2273 | 1480 |

Table 4: Average duration of training and epochs across models.

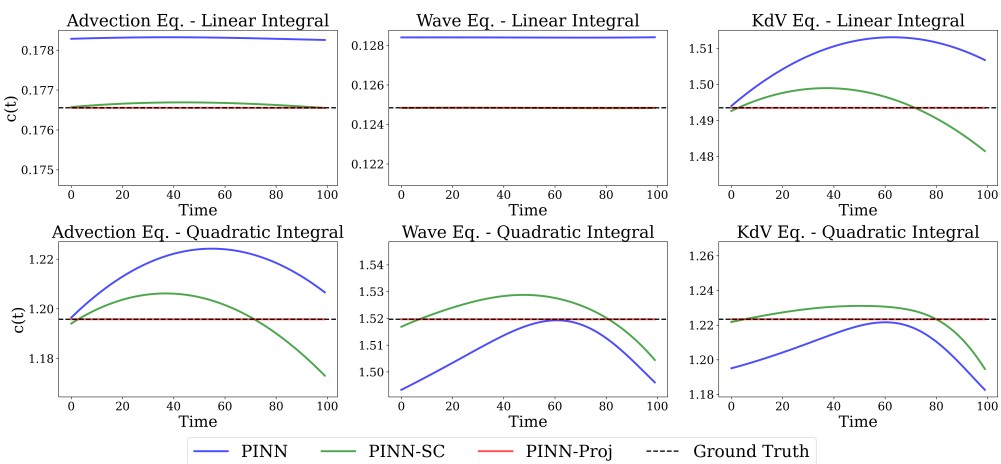

Figure 1: c(t) values over time for the conserved system PDEs with one conserved quantity

datasets by three to four orders of magnitude compared to the next best model, which was always PINN-SC.

Between different conserved quantities, PINN-SC always improved on PINN in Error $c$. In Error $u$, PINN-SC consistently improved over PINN in the Advection Equation and the Korteweg-De Vries Equation, but worsened it for the Reaction-Diffusion Equation. Error $c_L$ and Error $c_Q$ are also slightly higher for each PDE when both integrals are conserved compared to when only one is conserved.

Table 4 shows the average duration and number of epochs of training for PINN, PINN-SC, and PINN-Proj. The average was taken across trials with different conserved quantities, so there is a single value for each model and PDE. The soft constraint increased training time on average by 7.1x, and the projection method increased training time on average by 6.5x. On average, PINN-Proj took the fewest epochs to converge, and PINN-SC always took the most epochs to converge across all trials.

Figure 1 visualizes the evolution of $c(t)$ over time for the conserved system PDEs. The ground truth $c(t)$ is the black dashed line. We can observe that these predictions are consistent with the error values, as the predicted $c(t)$ values from PINN diverge most from the ground truth, those from PINN-SC diverge less, and those from PINN-Proj are difficult to discern from the ground truth on scales where all curves are visible.

## 5 DISCUSSION

On solution prediction, PINN-Proj achieved the best performance on 11 of the 15 systems tested. However, we note that the margin between PINN-Proj and the next best model was always less than

one order of magnitude. On conservation of quantities, PINN-Proj was always the best performing model, outperforming the soft constraint by three to four orders of magnitude. Our results thus demonstrate that the projection method was able to reliably and significantly improve the conservation of the linear and quadratic integral, both together and separately. Additionally, the projection method also marginally improved the accuracy of the PDE solution.

Error $c$ was consistently the lowest on both the Advection Equation, which is a first-order PDE, and higher on the Wave and Korteweg-De Vries Equations, which are second-order and third-order PDEs, respectively. This indicates that Error $c$ for PINN-Proj increased as the order of the PDE increased. Despite the Reaction-Diffusion Equation being a first order PDE, it still had Error $c$ values comparable to the higher order PDEs. This is likely due to the error of the linear approximation of $c(t)$, and could be reduced using higher order approximation methods. Error $c_L$ and $c_Q$ were also similar whether the linear and quadratic integrals were conserved at the same time or not, so conserving both quantities did not seem to significantly degrade performance.

**Errors and Guarantee**  Across all datasets and conserved quantities, the lowest Error $c$ values for PINN-Proj reached approximately 1E-6. This is about one order of magnitude away from machine precision for 32-bit floating point, at approximately 1E-7. However, we claim that our guarantee is validated, and that the discrepancy between these values originates from approximation error that we subsequently discuss in the solution data and model architecture. All numerically computed PDE solutions contain discretization error from the approximation of continuous derivatives and integrals on a finite computational grid. This error then accumulates at the integration step in the projection method and limits its accuracy. There is also numerical error from the solver method. We can estimate its contribution to Error $c$ by the change in $c(t)$ over time. Since $c(t)$ should remain constant in a conserved system, any variation must come from the numerical error. To address this, the mean $c(t)$ value was used as the ground truth of $c(t)$.

There is also approximation error inherent in the PINN architecture from the neural network and the auto-differentiation step in the PDE structure enforcement. These methods can have difficulty accurately representing solutions with sharp features and higher-order derivatives. The PINN is also limited by the set of collocation points that enforce the PDE structure. We observed that increasing the number of collocation points decreased Error $c$, as it allows the PINN to increase the resolution of its approximation of the PDE residual loss. Though these errors limit the application of the projection method, one could apply more sophisticated solver methods, increase the spatial resolution of the PDE solution and the collocation points, and increase the size of the neural network to further reduce conservation error and fully realize the benefits of the projection method.

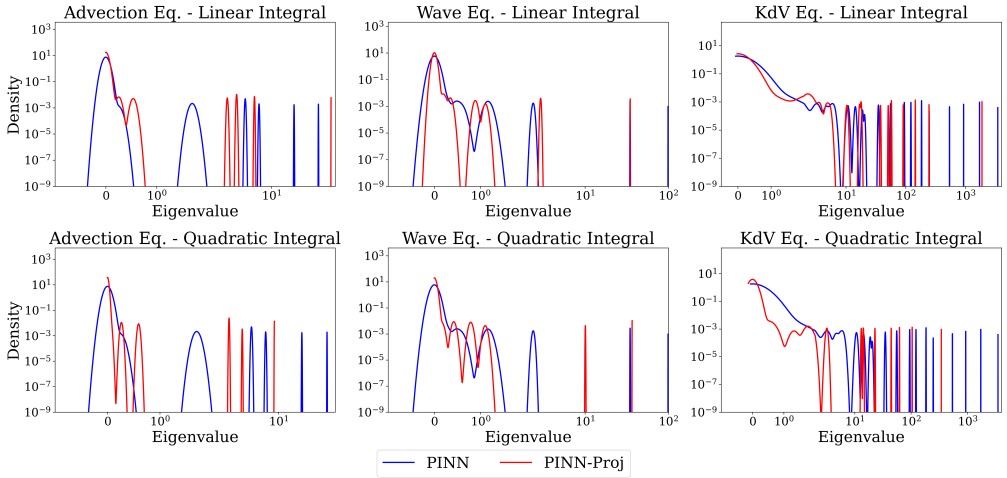

Figure 2: Spectral densities of Hessian for the conserved system PDEs with one conserved quantity

**PINN Training and Convergence**  Both the addition of the soft constraint in PINN-SC and the projection method in PINN-Proj significantly increased training time. This is because the projection method required the neural network $u_\theta(x, t)$ to be evaluated at all $x$ and $t$ values in the computational domain at each training step. Additionally, both the soft constraint and the projection layer also

require an additional summation calculation over the spatial points in the forward pass, which greatly increases the complexity of gradient computations during backpropagation. However, through this increased cost, we were able to guarantee the conservation of a chosen quantity and even improve the accuracy of the solution. For applications where safety is paramount, this increased cost could likely be justified. The cost is a one-time investment at training time, which can be amortized over all future inference uses. Additionally, in some applications such as in chemical reactors Chen et al. (2024) or inverse design Lu et al. (2021), hard constraints are necessary for safe and accurate modeling.

Though PINN-SC and PINN-Proj both added to training time, on average PINN-Proj always took the fewest epochs to converge. Combined with the improved Error $u$, the results suggest that the projection improved convergence. Since an accurate reconstruction of the solution would maintain the conserved quantity, the projection could have constrained the search space and guided gradient descent to a better solution more rapidly. We can verify this by analyzing the loss landscape using the spectral density of the loss function's Hessian matrix Rathore et al. (2024). These spectral densities describe the distribution of the eigenvalues of the Hessian matrix, and they give information on the local curvature of the loss function at the optimum found by gradient descent. When the eigenvalues of the Hessian are far apart in magnitude, different directions in the loss landscape vary in sensitivity to change, so the loss is ill-conditioned and difficult for gradient descent to minimize. When eigenvalues are similar in magnitude, there is more uniform sensitivity in all directions, so the loss landscape is better conditioned and gradient descent is more effective.

Figure 2 contains the spectral densities of the Hessians of PINN and PINN-Proj for the conserved systems. The spectral densities for PINN-Proj are more tightly clustered around 0 than PINN. Thus, in general, the projection lowered the magnitude of the eigenvalues. Additionally, the greatest eigenvalue for PINN-Proj always is lower in value than that of PINN. This suggests that the projection method improved the conditioning of the loss landscape, which could account for the improved convergence. However, it seemed that the projection did not improve the conditioning for the Advection Equation when the linear integral was conserved.

The spectral densities can also help understand to what degree the projection changed the conditioning of the loss landscape. By looking at how much more tightly clustered the eigenvalues are, we see that PINN-Proj improved the conditioning of the Korteweg-De Vries Equation the most, as it lowered the greatest eigenvalue by around a factor of 100 when the quadratic integral was conserved, and slightly worsened the conditioning of the Advection Equation when the linear integral was conserved. The Advection Equation was the simplest PDE tested, so its loss landscape would be relatively simple. The projection could, therefore, have had less of a benefit on the conditioning, as it was already well-conditioned. We discussed how the projection method also worsened some approximation errors, and this could have harmed the solution reconstruction. The other PDEs were, therefore, sufficiently complex such that the projection improved accuracy despite the accumulation of approximation errors.

Future work could involve extending the projection method to other integral quantities beyond the linear and quadratic integrals, which would involve solving a new non-linear optimization problem that is differentiable and tractable in neural network training. This direction is promising owing to the demonstrated effectiveness of the projection method. Future work could also include evaluating the projection method on more complex PDEs, including 2-dimensional PDEs, and initial conditions to further quantify its performance.

## 6 CONCLUSION

We proposed a novel projection method to guarantee conservation of integral quantities in a PINN. To determine its efficacy, we compared a PINN that uses the projection method, PINN-Proj, to an unmodified PINN and a PINN with a soft constraint, PINN-SC, on conserved and non-conserved PDE systems. We found that while adding computational cost, the addition of the projection significantly reduced the violation of conservation laws and guaranteed the conservation of the linear and quadratic integrals, both when applied separately and jointly, over PINN and PINN-SC. We also found that the projection improved the convergence of PINN using spectral analysis and marginally lowered the error in solution reconstruction.

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

## A APPENDIX

### A.1 TRAINING SETUP

The PINN models contained 9 hidden linear layers of 20 neurons. The weights of the neural network were initialized using Xavier normal initialization. All activation functions were hyperbolic tangent.

The optimizer was L-BFGS, and the stopping criteria was the change in the change in the loss function was less than machine epsilon, which was approximately 1E-16. Thus, the number of epochs of training varied between trials and between datasets. We used a weight of $\lambda = 10$ on PINN-SC. All training was done on an Nvidia RTX 4090.

### A.2 PDE Setup

Each PDE dataset had parameters $\Delta x = 1/128$, $\mathcal{X} = [0, 2]$, $\Delta t = 0.01$, $\mathcal{T} = [0, 0.99]$, $n_x = 256$, and $n_t = 100$, so each had 25,600 total state values $u$. We used figures for the conserved system PDEs to isolate the effects of the projection method without confounding factors from higher dimensionality or non-conservative terms.

The solutions for the Advection and Wave Equation were generated using a Weighted Essentially Non-Oscillatory (WENO) solver method. The solutions for the Korteweg-De Vries and Reaction-Diffusion Equation were generated using a Crank-Nicolson solver method. The Reaction-Diffusion equation had a diffusion coefficient of 0.1 and a reaction coefficient of 0.5. The Advection equation had a transport coefficient of 0.25. The Wave equation had a diffusion coefficient of 0.25. The Korteweg-De Vries had a non-linear coefficient of 1 and a dispersive coefficient of 0.0025.

### A.3 Hessian Spectral Densities Setup

The Hessian spectral densities were generated by stochastic Lanczos quadrature (SLQ) using PyHessian Yao et al. (2020). SLQ estimates the Hessian spectrum by using the Lanczos algorithm, which transforms a large matrix into a much smaller tridiagonal matrix whose eigenvalues approximate the original matrix's eigenvalues, with random probe vectors to iteratively construct a tridiagonal matrix whose eigenvalues approximate those of the full Hessian. The spectral densities were smoothed using Gaussian kernels with a variance of 4E-5.

### A.4 Advection Equation

The Advection Equation is a PDE that describes the transport of a quantity $u(x, t)$, such as mass, heat, or momentum, within a fluid due to its bulk motion.

$$\frac{\partial u}{\partial t} + c\frac{\partial u}{\partial x} = 0$$

Initial Condition:

$$u_0 = \exp\left(-\frac{(x-1)^2}{(0.1)^2}\right)$$

Boundary Condition:

$$\frac{\partial u}{\partial x}(0, t) = 0, \quad \frac{\partial u}{\partial x}(2, t) = 0$$

### A.5 Wave Equation

The Wave Equation is a second-order PDE that describes the propagation of waves where a quantity $u(x, t)$ is the wave's displacement.

$$\frac{\partial^2 u}{\partial t^2} = c^2\frac{\partial^2 u}{\partial x^2}$$

Initial Condition:

$$u_0 = \exp\left(-(x-1)^2\right)$$

Boundary Condition:

$$\frac{\partial u}{\partial x}(0, t) = 0, \quad \frac{\partial u}{\partial x}(2, t) = 0$$

### A.6 KORTEWEG-DE VRIES EQUATION

The Korteweg-de Vries equation is a third-order non-linear PDE that describes the propagation of shallow water waves where a quantity $u(x,t)$ represents the wave's amplitude.

$$\frac{\partial u}{\partial t} + au\frac{\partial u}{\partial x} + b\frac{\partial^3 u}{\partial x^3} = 0$$

Initial Condition:
$$u_0 = \exp\left(-(x-1)^2\right)$$

Boundary Condition:
$$\frac{\partial u}{\partial x}(0,t) = 0, \quad \frac{\partial u}{\partial x}(2,t) = 0$$

### A.7 REACTION-DIFFUSION EQUATION

The Reaction-Diffusion Equation models the interaction between diffusion, which spreads a quantity over space, and reaction, which governs the flux into the system where a quantity $u(x,t)$ is the concentration of a substance.
$$\frac{\partial u}{\partial t} = D\frac{\partial^2 u}{\partial x^2} + R(u)$$

In this case,
$$R(u) = ku.$$

Initial Condition:
$$u_0 = \exp\left(-\frac{(x-1)^2}{(0.5)^2}\right)$$

Boundary Condition:
$$\frac{\partial u}{\partial x}(0,t) = 0, \quad \frac{\partial u}{\partial x}(2,t) = 0$$

### A.8 LINEAR APPROXIMATION OF $c(t)$

Second-order central difference:
$$c'(t_i) = \frac{c(t_{i+1}) - c(t_{i-1})}{t_{i+1} - t_{i-1}}, \tag{19}$$

with forward differences at the left boundary:
$$c'(t_0) = \frac{-3c(t_0) + 4c(t_1) - c(t_2)}{t_2 - t_0}, \tag{20}$$

and backward differences at the right boundary:
$$c'(t_n) = \frac{3c(t_n) - 4c(t_{n-1}) + c(t_{n-2})}{t_n - t_{n-2}}. \tag{21}$$

We approximate $c(t_f)$ using gradient-based interpolation, where $t^*$ is the closest value to $t_f$ where $c(t)$ is defined:
$$c(t_f) = c(t^*) + c'(t^*)(t_f - t^*). \tag{22}$$

### A.9 DERIVATION OF CONSERVATION OF LINEAR INTEGRAL

$$\min_y ||\bar{u}_\theta(x,t) - y||^2, \text{ s.t. } \Delta x \mathbf{1}_n^\top y = c(t) \tag{23}$$

The Lagrangian has the form:
$$\mathcal{L}(y, \lambda) = (\bar{u}_\theta(x,t) - y)^\top (\bar{u}_\theta(x,t) - y) + \lambda(\Delta x \mathbf{1}_n^\top y - c(t)). \tag{24}$$

Take partial derivatives and set to zero:

$$\partial \mathcal{L} / \partial y_i = -2(\bar{u}_\theta(x_i, t) - y_i) + \lambda \Delta x = 0 \tag{25}$$

$$\partial \mathcal{L} / \partial \lambda = \Delta x \mathbf{1}_n^\top y - c(t) = 0 \tag{26}$$

From the first equation, we get:

$$y_i = \bar{u}_\theta(x_i, t) - \frac{\lambda \Delta x}{2} \tag{27}$$

Substitute this into the second equation:

$$c(t) = \sum_{i=1}^n y_i \Delta x = \sum_{i=1}^n (\bar{u}_\theta(x_i, t) - \frac{\lambda \Delta x}{2}) \Delta x \tag{28}$$

$$\frac{\lambda \Delta x}{2} = \sum_{i=1}^n \frac{\bar{u}_\theta(x_i, t)}{n} - \frac{c(t)}{n \Delta x} \tag{29}$$

Substitute back into the equation for $y_i$ to find the projection solution on the discretized point:

$$\tilde{\bar{u}}_\theta(x_i, t) = y_i = \bar{u}_\theta(x_i, t) + \frac{c(t)}{n \Delta x} - \sum_{i=1}^n \frac{\bar{u}_\theta(x_i, t)}{n} \tag{30}$$

$$= \bar{u}_\theta(x_i, t) + \frac{c(t)}{n \Delta x} - \mathbf{1}_n^\top \frac{\bar{u}_\theta(x_i, t)}{n} \tag{31}$$

We can verify that this solution satisfies the constraint:

$$\sum_{i=1}^n \tilde{\bar{u}}_\theta(x_i, t) \Delta x = \sum_{i=1}^n (\bar{u}_\theta(x_i, t) + \frac{c(t)}{n \Delta x} \tag{32}$$

$$- \sum_{i=1}^n \frac{\bar{u}_\theta(x_i, t)}{n}) \Delta x = c(t)$$

Extending the projection to continuous space yields:

$$\tilde{u}_\theta(x, t) = u_\theta(x, t) + \frac{c(t)}{n \Delta x} - \sum_{i=1}^n \frac{\bar{u}_\theta(x_i, t)}{n} \tag{33}$$

### A.10 Derivation of Conservation of Quadratic Integral

$$\min_y ||\bar{u}_\theta(x, t) - y||^2, \text{ s.t. } \Delta x y^\top y = c(t) \tag{34}$$

The Lagrangian has the form:

$$\mathcal{L}(y, \lambda) = (\bar{u}_\theta(x, t) - y)^\top (\bar{u}_\theta(x, y) - y) \tag{35}$$

$$+ \lambda (\Delta x y^\top y - c(t))$$

Take partial derivatives and set to zero:

$$\partial \mathcal{L} / \partial y_i = -2(\bar{u}_\theta(x_i, t) - y_i) + 2\lambda \Delta x y_i = 0 \tag{36}$$

$$\partial \mathcal{L} / \partial \lambda = \Delta x y^\top y - c(t) = 0 \tag{37}$$

From the first equation, we get:

$$y_i = \frac{\bar{u}_\theta(x_i, t)}{1 + \lambda \Delta x} \tag{38}$$

Substitute this into the second equation:

$$c(t) = \sum_{i=1}^n y_i^2 \Delta x = \sum_{i=1}^n \frac{\bar{u}_\theta(x_i, t)^2 \Delta x}{(1 + \lambda \Delta x)^2} \tag{39}$$

$$\frac{1}{1 + \lambda \Delta x} = \sqrt{\frac{c(t)}{\sum_{i=1}^n \bar{u}_\theta(x_i, t)^2}} \tag{40}$$

On the dicretized point, the projected solution that satisfies the constraint is:

$$\tilde{\bar{u}}_\theta(x_i, t) = y_i = \bar{u}_\theta(x_i, t)\sqrt{\frac{c(t)}{\sum_{i=1}^{n} \bar{u}_\theta(x_i, t)^2}} \tag{41}$$

Extending the projection to continuous space yields:

$$\tilde{u}_\theta(x, t) = u_\theta(x, t)\sqrt{\frac{c(t)}{\sum_{i=1}^{n} \bar{u}_\theta(x_i, t)^2}} \tag{42}$$

### A.11 Derivation of Conservation of Combined Linear and Quadratic Integral

$$\min_y ||\bar{u}_\theta(x, t) - y||^2 \tag{43}$$

$$\text{s.t. } \Delta x y^\top y = c_1(t) \tag{44}$$

$$\Delta x \mathbf{1}_n^\top y = c_2(t)$$

The Lagrangian has the form:

$$\mathcal{L}(y, \lambda) = (\bar{u}_\theta(x, t) - y)^\top (\bar{u}_\theta(x, y) - y) \tag{45}$$

$$+ \lambda_1 (\Delta x y^\top y - c_1(t))$$

$$+ \lambda_2 (\Delta x \mathbf{1}_n^\top y - c_2(t))$$

Take partial derivatives and set to zero:

$$\partial\mathcal{L}/\partial y_i = -2(u_\theta(x_i, t) - y_i) \tag{46}$$

$$+ 2\lambda_1 \Delta x y_i + \lambda_2 \Delta x = 0$$

$$\partial\mathcal{L}/\partial\lambda_1 = \Delta x y^\top y - c_1(t) = 0 \tag{47}$$

$$\partial\mathcal{L}/\partial\lambda_2 = \Delta x \mathbf{1}_n^\top y - c_2(t) = 0 \tag{48}$$

From the first equation, we get:

$$y_i = \frac{2u_\theta(x_i, t) - \lambda_2 \Delta x}{2 - 2\lambda_1 \Delta x} \tag{49}$$

Substitute into constraint:

$$c_1(t) = \Delta x \sum_{i=1}^{n} \frac{(2u_\theta(x_i, t) - \lambda_2 \Delta x)^2}{(2 - 2\lambda_1 \Delta x)^2} \tag{50}$$

$$c_2(t) = \Delta x \sum_{i=1}^{n} \frac{2u_\theta(x_i, t) - \lambda_2 \Delta x}{2 - 2\lambda_1 \Delta x} \tag{51}$$

From second equation:

$$\lambda_2 \Delta x = \frac{-(2 - 2\lambda_1 \Delta x)c_2(t)}{n\Delta x} + \sum_{i=1}^{n} \frac{2u_\theta(x_i, t)}{n} \tag{52}$$

$$y_i = \frac{2u_\theta(x_i, t) - \lambda_2 \Delta x}{2 - 2\lambda_1 \Delta x} \tag{53}$$

$$= \frac{u_\theta(x_i, t) - \sum_{i=1}^{n} \frac{u_\theta(x_i, t)}{n}}{1 - \lambda_1 \Delta x} + \frac{c_2(t)}{n\Delta x}$$

Let $\bar{u}_\theta = \sum_{i=1}^{n} \frac{u_\theta(x_i, t)}{n}$ and substitute the above into the first equation:

$$= \Delta x \sum_{i=1}^{n} \left( \frac{(u_\theta(x_i, t) - \bar{u}_\theta)^2}{(1 - \lambda_1 \Delta x)^2} + \right. \tag{54}$$

$$\left. 2\frac{u_\theta(x_i, t) - \bar{u}_\theta}{1 - \lambda_1 \Delta x} \frac{c_2(t)}{n\Delta x} + \frac{c_2^2(t)}{(n\Delta x)^2} \right)$$

$u_\theta(x_i, t) - \overline{u}_\theta$ cancels out after the summation, so the middle term is 0

$$c_1(t) = \Delta x \sum_{i=1}^{n} \left( \frac{(u_\theta(x_i, t) - \overline{u}_\theta)^2}{(1 - \lambda_1 \Delta x)^2} + \frac{c_2^2(t)}{(n\Delta x)^2} \right) \tag{55}$$

$$\frac{c_1(t)}{\Delta x} - \frac{c_2^2(t)}{n\Delta x^2} = \frac{1}{(1 - \lambda_1 \Delta x)^2} \sum_{i=1}^{n} (u_\theta(x_i, t) - \overline{u}_\theta)^2 \tag{56}$$

$$\frac{1}{1 - \lambda_1 \Delta x} = \sqrt{\frac{c_1(t)/\Delta x - c_2^2(t)/(n\Delta x^2)}{\sum_{i=1}^{n} (u_\theta(x_i, t) - \overline{u}_\theta)^2}} \tag{57}$$

Substitute back to the expression of $y_i$

$$y_i = (u_\theta(x_i, t) - \overline{u}_\theta) \sqrt{\frac{c_1(t)/\Delta x - c_2^2(t)/(n\Delta x^2)}{\sum_{i=1}^{n} (u_\theta(x_i, t) - \overline{u}_\theta)^2}} + \frac{c_2(t)}{n\Delta x} \tag{58}$$

### A.12   LLM USE

LLMs were used to aid in data imputation, improve the wording of the paper, and help write code. High level planning and most code writing was carried out by humans, but some repetitive code and specifics in creating the solvers were aided by LLMs.

