# OpenReview forum: "Guaranteeing Conservation of Integrals with Projection in Physics-Informed Neural Networks"
_ICLR.cc/2026/Conference — Submitted to ICLR 2026_

### Official Review · Reviewer_fZpm · 2025-10-30

**Soundness:** 3
**Presentation:** 3
**Contribution:** 3
**Rating:** 4
**Confidence:** 4

**Summary:**

This paper studies the projection method for the conservation of integral quantities in PINNs. It can be regarded as additional regularization for solutions of PINNs, with additional knowledge on integral quantities (energy). Instead of adopting soft constraints in the loss, the paper formulates it as a nonlinear projection after the training of PINNs, therefore, acting as hard constraints. Experiments show the superiority of the proposed projected PINNs over the soft-constrained PINNs.

**Strengths:**

(1) The presentation is good.

(2) The method is simple and motivated.

(3) The conservation of solution energy boosts the performance of PINNs.

(4) The experimental results are strong that PINN-Proj consistently outperforms PINN and PINN-SC.

**Weaknesses:**

(1) The scale of the method (for high-dimensional PDE) is difficult. The advantages of PINNs over classical numerical methods are their potential in high-dimensional PDEs. However, the projection method still suffers from the curse of dimensionality, since it requires the discretization of the space. Therefore, the nonlinear projection is infeasible and computationally expensive for high-dimensional PDEs. I think it is the main weakness of the method. PINNs try to avoid discretization, but the proposed method requires it.

(2) It is good if you can present the algorithm, so that readers can easily capture the core idea of this paper.

**Questions:**

The projection applies once after the training of PINNs. For example, if PINNs overfit the data so that it does not satisfy the conservation law, will the projection further degrade the performance of PINNs? I think if we can apply the projection during PINNs training (alternating between PINNs training and conservation projection), then the projection will gradually guide the PINNs to learn this conservation.

---

### Official Review · Reviewer_xnP5 · 2025-10-31

**Soundness:** 1
**Presentation:** 2
**Contribution:** 1
**Rating:** 2
**Confidence:** 3

**Summary:**

This paper proposes a projection-based method to enforce conservation laws within Physics-Informed Neural Networks (PINNs) by incorporating linear and quadratic integral constraints through a post-training projection step. The approach aims to improve the physical fidelity of PINN solutions by ensuring that conserved quantities are satisfied exactly, thereby addressing common issues with conservation violations in neural PDE solvers. The authors demonstrate the method on several benchmark PDEs, showing improved conservation and, in some cases, enhanced convergence properties.

**Strengths:**

The paper addresses an important limitation of PINNs by proposing a projection method that guarantees the conservation of integral quantities. The formulation is mathematically well-motivated and provides a clear, closed-form projection for enforcing conservation for 1D uniform-grid cases. Empirical results demonstrate consistent improvements on standard benchmark problems.

**Weaknesses:**

1. It is unclear whether the observed improvement in PDE solution error is truly due to the enforcement of conservation constraints. The error reduction may result either from matching the integral quantities or from the projection improving training stability. An ablation study is needed to clarify this.
2.  The post-hoc projection does not consider the PDE residual. Therefore, even if the conservation constraint is enforced, the solution may violate fundamental physical laws, such as mass, momentum, or energy conservation, especially in nonlinear or strongly coupled systems. This raises concerns about the physical plausibility of the projected solutions.
3. It is unclear whether the conservation is preserved at points $(x,t)$ outside the training set. Demonstrating generalization beyond training data—particularly in comparison with approaches like PINN-SC—is necessary.
4. The comparison set is limited. Evaluating only vanilla PINN and soft-constrained PINN may be insufficient. In particular, PINN-SC requires careful tuning of the regularization parameter. Additional baselines, such as Lagrange multiplier-based methods, would strengthen the evaluation.
5. Since the integral is approximated via a discretized sum, the projection inherits discretization errors. The method may be sensitive to the number of discretization points $n$. Moreover, the method’s scalability to higher-dimensional domains (e.g., 2D or 3D) is questionable because:
* The number of constraints grows significantly,


* The computational cost of the projection becomes prohibitive even in 1D,


* Closed-form projection solutions may not exist.
Furthermore, extending this approach to tensor- or vector-valued fields is nontrivial.

Therefore, experiments on higher-dimensional problems are essential to validate the method’s scalability and practical feasibility. Extending the projection to non-uniform grids, adaptive meshes, or higher-dimensional domains (>=3D) may not be straightforward, and closed-form solutions may not exist for such cases.

6. The experiments are limited to simple, low-dimensional examples. It remains unclear how much enforcing conservation alone improves PDE accuracy or stability, and whether the approach generalizes to a broader class of PDEs or practical problems.

7. The claim that Hessian eigenvalues of PINN-Proj are more tightly clustered near zero, suggesting improved conditioning, is potentially misleading. Eigenvalues concentrated near zero typically indicate ill-conditioning, which can slow or destabilize optimization. A more detailed explanation and clarification are necessary. Additionally, eigenvalue distributions alone may not fully characterize optimization performance or convergence, so such claims should be presented cautiously.

8. For hyperbolic conservation laws, convergence to the entropy solution is crucial before enforcing conserved quantities. The proposed method does not address this requirement, which may limit its applicability to such systems.



**Minor Comments**

* Subsections 3.1 and 3.2 do not present the proposed methodology of this paper and would be better placed in a preliminary or background section for clarity.

* In Eq. (4), the conserved quantity c depends on the solution $u$. Therefore, it would be more precise to denote it as $c(u(⋅,t))$ or $c[u](t)$ instead of simply $c(t)$, to explicitly indicate this dependency.

**Questions:**

Please refer to the comments in the Weaknesses section for detailed remarks.

---

### Official Review · Reviewer_3b9M · 2025-10-31

**Soundness:** 3
**Presentation:** 2
**Contribution:** 2
**Rating:** 2
**Confidence:** 4

**Summary:**

The manuscript considers physics-informed neural networks (PINNs) for the solution of evolution equations with conserved quantities. It proposes a method of obtaining predictions that satisfy the conservation law exactly based on an explicit transformation of the network's output. The resulting method evalauted on a variety of PDEs showing improved accuracy and conservation of the quantities over vanilla PINNs.

**Strengths:**

+ The manuscript treats the imporant and timely problem of producing more physically plausible predictions with neural networks.
+ The manuscript gives a clean way to exactly impose conservation of linear and quadratic quantities.

**Weaknesses:**

+ Depth of the contribution: The manuscript proposes essentially a post-processing of the predictions. Whereas a slight improvement over vanilla PINNs is shown, it is not clear, whether this enables the application of PINNs to problems previously out of reach.
+ Ablation: The post-processing is used during training. An ablation study of training without the post-processing, but using it at evaluation is not presented.
+ The Complexity of the training procedure and training times are not discussed and reported.
+ Evaluation:
	+ The manuscript does not give insight into the training process. In particular, the choice of the optimizer is crucial in PINNs as demonstrated by the recent success of second-order optimizers like energy natural gradient. In particular, energy natural gradient leads to a loss with an optimal condition number.
	+ Results not competitive with PINNs optimized with current state-of-the-art optimizers.
+ The method is relatively ad-hoc, hence only applicable to the mean and variance as conserved quantities.
+ At times, the manuscript is not an entirely smooth read, e.g., in the abstract *While the soft constraint PINNs use to enforce the structure of partial differential equations (PDEs) enables necessary flexibility during training* might be a typo? Also, in Subsection 3.1 it is not clear what is meant by *The physics-informed neural network (PINN) is defined as $f\coloneqq u_t + \mathcal N[u]$*_. Firstly, it is good to remind the reader that $u_t$ refers to the time derivative. Secondly, it is a bit of a stretch to call the residual $f$ _physics-informed neural network_. Further, Section 3 consists of a lot of very short subsections and would benefit from streamlining and condensation.

**Questions:**

+ Which optimizer was used during training?
+ How does your architecture perform when compared to PINNs trained with second-order optimizers?
+ Can you use second-order optimizers to efficiently train your architecture?
+ Can you generalize your method to other conservation laws?

---

### Official Review · Reviewer_wwvk · 2025-11-01

**Soundness:** 1
**Presentation:** 3
**Contribution:** 2
**Rating:** 2
**Confidence:** 4

**Summary:**

This paper proposed an analytical formula for enforcing conservation law in the form of $u_{t}=-\nabla \cdot \mathbf{F}(u)$. The integral form  $\frac{d}{dt} c(t)=\frac{d}{dt} \int_{\mathcal{X}} u\; dx=-\oint \mathbf{F}(u) \cdot d \mathbf{S}$ describe how the conserved quantity $c(t)$ changes overtime, and $c(t)$ can be computed via temporal integration. The key idea is to represent the PINN ansatz $u_{\theta}$ by its evaluation on an uniform grid on the domain, and compute the conserved quantity using an uniform quadrature: $\hat{c}(t)\approx 1_{n}^{\top}u_{\theta} \Delta x$. The constraint $\hat{c}(t)=c(t)$ becomes linear, and a minimum L2 distantce projection $\tilde{u}_{\theta}$ has analytical solution which depends on $c(t)$. The method is tested on 4 1D equations and show that conservation law constraint is well-satisfied by the method.

**Strengths:**

The paper is well-written and is easy to follow. The proposed method is mathematically clear and easy to implement.

**Weaknesses:**

- One of the key strengths of PINN is its mesh-free property. Enforcing the constraint by projecting PINN to a uniform grid defeats the purpose of having PINN in the first place. If we are already projecting PINN to a uniform grid, why don't we use a finite element? Also, from Table 4, we see that PINN-proj is very slow, exactly due to the use of a uniform mesh. Also, this is just 1D; for 2D and 3D, it would be much worse. Furthermore, in real problems, the domain could be irregular, and uniform discretization doesn't always work. The paper didn't discuss these limitations at all.
- The baseline is weak: one can use augmented Lagrangian, ADMM, and even just a straightforward soft constraint $\lambda |\mathbb{E}[u_{\theta}] - c(t)|^{2}$. Furthermore, the baseline PINN-SC uses only a single weight of $\lambda=10$, without any ablation.
- The experiment in the paper is only toy examples. Showing the method works on a more realistic example would greatly increase the impact.

**Questions:**

- Why is the extending the projection to continuous space (eq. 33) valid?
- How is $c(t)$ calculated?
- The initial condition doesn't satisfies the boundary condition. Why is that the case?
- Although the paper proposed a method that works for $\frac{d}{dt}c(t)\neq 0$, for example the Reaction-Diffusion, Figure 1 only show the case for $\frac{d}{dt}c(t)= 0$. Why is that the case? Non constant $c(t)$ is much more interesting.

---

### Meta-Review · Area_Chair_GVeE · 2026-01-06

**Summary:**

Taken together, the reviews indicate a clear negative consensus: the paper’s contributions are judged incremental and the validation does not yet meet the level of breadth and rigor expected for acceptance. So my recommendation is reject.

**Reviewer Concerns:**

The main sticking points are repeatedly raised across reviews: (a) the paper needs a sharper comparison and narrative against the closest related methods to establish incremental vs. substantive contribution; (b) the empirical section would benefit from broader and more controlled experiments (e.g., stronger contemporary baselines, multiple seeds, fair hyperparameter parity, and more challenging/realistic settings); and (c) more diagnostic evidence is needed.

**Reviewer Scores:**

The scores remain consistently below threshold, and the discussion does not indicate that the decisive concerns have been resolved in a way that would justify an upward shift; the overall recommendation therefore remains reject.

---

### Decision · Program_Chairs · 2026-01-26

Reject